# Exploring the Impact of Mining on Community Health and Health Service Delivery: Perceptions of Key Informants Involved in Gold Mining Communities in Burkina Faso

**DOI:** 10.3390/ijerph20247167

**Published:** 2023-12-12

**Authors:** Gianna S. Himmelsbach, Hyacinthe R. Zabré, Andrea Leuenberger, Astrid M. Knoblauch, Fritz Brugger, Mirko S. Winkler

**Affiliations:** 1Swiss Tropical and Public Health Institute, 4123 Allschwil, Switzerland; gianna.himmelsbach@datazug.ch (G.S.H.); astrid.knoblauch@unibas.ch (A.M.K.); 2University of Basel, P.O. Box 4001 Basel, Switzerland; 3Africa Centres for Disease Control and Prevention, African Union Commission, Roosevelt Street W21 K19, Addis Ababa 3243, Ethiopia; zraogohyacinthe@yahoo.fr; 4Swiss Red Cross, 3001 Bern, Switzerland; 5ETH Zürich, Swiss Federal Institute of Technology, 8092 Zurich, Switzerland; fritz.brugger@nadel.ethz.ch

**Keywords:** Burkina Faso, health determinants, health effects, industrial mining, qualitative research

## Abstract

Sub-Saharan Africa is rich in natural resources but also faces widespread poverty. The United Nations’ Sustainable Development Goals brought increased attention to resource extraction projects, emphasizing their development potential in extraction regions. While mining companies are required to conduct environmental impact assessments, their effect on the project-affected communities’ health mostly lacks systematic management, and their consideration of community perspectives is insufficient. Between March and May 2019, qualitative research was conducted at three industrial gold mines in Burkina Faso. Thirty-six participants, including community leaders, healthcare providers, and mining officials, were interviewed through key informant interviews about their perceptions on the impacts of mining operations on health, health determinants, and health service delivery. Disparities in perceptions were a key focus of the analysis. Mining officials reported mainly positive effects, while healthcare providers and community leaders described enhancing and adverse health impacts without clear trends observed regarding the extent of the impacts on health determinants. The perception of predominantly positive health impacts by mining officials represents a potential risk for insufficient acknowledgement of stakeholders’ concerns and mining-related effects on community health in affected populations. Overall, this study enhances comprehension of the complex interplay between mining operations and health, emphasizing the need for comprehensive assessments, stakeholder involvement, and sustainable practices to mitigate negative impacts and promote the well-being of mining communities.

## 1. Introduction

Sub-Saharan Africa (SSA) is well-known for its abundance of natural resources but also for the fact that its countries are among the poorest in the world [1]. With the adoption of the 17 Sustainable Development Goals (SDGs) by the United Nations in 2015, natural resource extraction projects (NREPs) have gained increased attention regarding their potential to influence various SDGs [1,2,3,4,5]. Often, SDG 3, that is, “ensure healthy lives and promote well-being for all at all ages”, is left behind by mining companies despite its importance for sustainable development and NREPs impacts on affected communities’ health [6]. Unlike with the environmental impact of NERPs, there is no regulatory obligation in low-income countries (LICs) to address the public health impact of NREP systematically, through health impact assessments (HIA) or otherwise [7]. Companies also do not or only marginally incorporate public health in their impact assessments on a voluntary basis [8,9].

Several adverse effects on the health of mine workers [10] and the population surrounding mining sites have been described in the literature [11,12,13]. Examples of destructive health impacts include elevated rates of HIV/AIDS [14], endemic diseases such as malaria [15], mental health problems [16,17], and conditions triggered by environmental impacts such as toxic chemicals [18,19,20,21]. In addition, NREP sites often experience a high influx of migrants due to the creation of new jobs and secondary businesses [22,23]. Because of the described health impacts above, mine operations can also affect local health services. An increase in patients and the appearance of new or unknown diseases can usher the local health systems into capacity problems. Consequently, these impediments to delivering quality care can have additional adverse effects on the health status of local communities. However, studies investigating the implications of industrial mining for the delivery of health services are rare.

Besides the negative impacts, NREPs can potentially improve population health [24,25]. Often, NREPs have their own health infrastructure for their employees, ensuring that the burden of treating injured or sick workers is not passed on to the public health centers [26]. In addition, investments in public infrastructure such as schools, water supply, or health centers address important social determinants of health (SDH) which benefit local communities directly [27,28]. Such investments typically take the form of voluntary corporate social responsibility (CSR) strategies, which are often found to be designed in a functionalist way. This means that companies prioritize operational needs and measures that smoothen operations when deciding on community investment. This can include measures that help win community leaders’ consent or diffuse tensions and opposition against mining. CSR is a function of improving operations in the first place. Such measures can also positively affect public health but are not designed from a public health perspective. As a result, they often have short-term effects but do not necessarily produce systemic and lasting solutions [29,30,31,32,33,34].

Although impacted communities appreciate some of these beneficial interventions, they still perceive many negative impacts and criticize the fact that interventions do not meet their expectations [35,36]. Meanwhile, mining officials appear to reject concerns expressed by affected communities, for example, about adverse health effects from dust or noise pollution or adverse effects on quality of life [37]. Disparities in expectations and perceptions can lead to misunderstandings, mistrust, or even conflicts [35]. Overcoming the functionalist approach to community (health) investment requires mining companies to involve project-affected communities in situation analysis and in decision-making, as proposed by the HIA guidelines [27,38]. Communities exposed to existing mining projects have accumulated experiences with mining-related health impacts and may perceive other interventions as helpful compared to mining companies’ representatives. Community involvement can lead to less tension and conflicts and help develop effective strategies for addressing health risks.

Systematic reviews, however, identified a lack of qualitative studies exploring the perceptions of local communities, local leaders, and health workers regarding mining-induced health impacts [39,40,41]. Furthermore, it is not well understood how the mining industry affects the health sector and, ultimately, healthcare delivery in project areas. Existing publications deal predominantly with mine workers’ health, neglect impact pathways focusing on SDH, and study LICs the least [39]. Therefore, research should shift its focus to a more holistic approach that incorporates the health experience of surrounding communities and particularly vulnerable groups such as children, adolescents, pregnant women, or residents from rural areas.

Our study contributes to reducing these gaps by exploring the perceived impacts of industrial gold mining on the health and health determinants of affected communities and healthcare service provision by different affected actors of the mine, the health sector, and the local communities. The following questions guided the research: (i) What are the perceived impacts of mining on health, selected determinants of health, and healthcare delivery? (ii) How do these perceived impacts differ among different actors?

## 2. Materials and Methods

This qualitative study was carried out as part of the health impact assessment for the sustainable development (HIA4SD) project, which has the overarching goal of informing a policy dialogue for promoting the application of HIA in SSA. The detailed project description can be found elsewhere [5,42]. In the area of health, qualitative research has been demonstrated to be considerably beneficial [43]. For the study presented, key informant interviews (KIIs) were conducted at three gold mining areas in Burkina Faso that served to explore disparities in perceptions of health impacts caused by mining operations. To uncover reasons behind certain community behaviors and local attitudes, KIIs with stakeholders from project-affected communities were conducted within the frame of the overarching research project using the COREQ criteria as guidance for this qualitative research [42,44].

### 2.1. Study Sites

The data used for this study were collected around three gold mines in Burkina Faso: (1) Bissa Gold Mine in Kongoussi health district; (2) Houndé Gold Mine in Houndé health district; and (3) Yaramoko Gold Mine in Bagassi health district (Figure 1; a short description of each mining site is provided in Appendix A). The Houndé Mine and the Bissa Mine are open pit mines, whereas the Yaramoko Mine is an underground mine. Extractions at each of the three mines only started a few years prior to data collection; their expected lifetimes range between 10 to 21 years [45,46,47]. Table 1 compiles the diverse features of the three mining projects.

### 2.2. Sampling and Recruitment of Study Participants

Prior to any field work, officials at the provincial or regional level and the management of the mining projects in the study areas were contacted through letters, phone calls, or direct visits in the field. Subsequently, the field team, comprising individuals of various genders, identified one to three local gatekeepers per site. Gatekeepers are individuals from the community who are very knowledgeable about the area and project development. The gatekeepers accompanied the field workers during the so-called transect walks and introduced them to local communities, local leaders, and the local health sector. Previous studies in rural areas successfully used transect walks to systemically identify relevant infrastructures and services for local communities and establish a first relationship with residents and the study area [48,49,50,51]. A more detailed description of the transect walk methodology is provided elsewhere [52].

Research participant selection was conducted by purposive sampling [53,54] of the major stakeholders, which included mining officials, healthcare providers (all from public healthcare facilities), and community leaders. Respondents were required to have resided or served in the community for at least one year. If multiple potential respondents in one stakeholder group were identified, the one who lived longest in the community was chosen to be interviewed. The targeted sample size was 6 people per stakeholder group, adding up to 18 people per study site. Thus, the overall targeted sample size was 54. It must be noted here that no one from the Bissa Gold Mine and only one person from the Houndé Gold Mine management participated in the study.

### 2.3. Data Collection

Data were collected between March and May of 2019, but preparations for the field work already started 2 to 4 weeks prior to arrival in the field. Each KII was conducted by one to two field team members (male). A semi-structured interview guide served as the basis for the interviews. Each KII comprised of two separate parts: Part A contained open-ended questions to explore the health impacts of the large-scale mining ongoing in the community and the health system response to these impacts as perceived by the three different stakeholder groups (i.e., mining officials, healthcare providers and community leaders). Part B required the interviewees to rate selected factors concerning health and health service delivery. The first question asked the participants to rate selected health determinants which are commonly affected by NREPs as having “improved”, “had no effect”, “had no effect”, or “worsened”. Based on previous research studying how NREPs affect community health [8,28,55], the selected health determinants were “Drinking water”, “Sanitation”, “Income levels”, “Employment”, “Road Network”, “Electricity”, “Educational institutions”, and “Women’s empowerment”. Respondents answered verbally and had to reason their answers. The following two questions required participants to assess the overall impact of mining on the health of the community and on the delivery of health services on a 5-point Likert scale, where 1 means “Strongly beneficial” and 5 means “Very negative”. The KIIs were carried out at the interviewees’ offices or at a location of the interviewees’ preference where adequate privacy and minimal distraction was assured. Before starting the interview, background information on the participant was collected. The majority of the KIIs were conducted in the country’s official language (French), while some were held in the local language (Mooré in Kongoussi, Dioula in Houndé, and Bagassi). The interviewers were responsible for documenting the session using the voice-recorder and by taking notes. After a day of field activities, debriefing sessions were held among the field team to discuss the progress of data collection and relay the most important findings.

#### Quality Assurance

The quality of the data collected was guaranteed by prior training of the field team and a project-specific data collection manual to which they could refer at any time in case of uncertainties about tools or procedures. Furthermore, the field coordinator performed daily debriefings with the field team. Such debriefings have proven helpful to ensure a harmonized data collection and that all interviewers have a sound understanding of perceived impacts in the study area [56,57].

### 2.4. Data Management and Analysis

Research assistants transcribed and translated, if necessary, the KIIs into French. The collected qualitative data were analyzed by the first author using thematic coding. Respondents’ answers were coded according to a previously defined codebook to identify concepts which were grouped into categories such as “Health Mining Effects” or “Health Service Provision Mining Effects”. Where needed, additional codes were added during the analysis (e.g., Health Mining Effects_NoEffect or Own Health Facilities_Mine). Perceived impacts on healthcare delivery were further categorized as service availability, general service readiness and service-specific readiness. These sub-codes are based on the WHO’s Service Availability and Readiness Assessment (SARA) survey, designed to assess the performance of health service delivery [58]. However, the categorization was adapted to this study and only superficial reports on tracer indicators used for SARA.

A qualitative data analysis software (Nvivo 13, QSR International, Victoria, Australia) was used for coding. Concept maps, word trees, or coding queries helped to gain a better understanding of the data. The answers of the factor ratings (Part B) were summarized and visualized using Microsoft Excel (2021, Microsoft, Redmond WA, USA). Noteworthy differences in the ratings were further investigated by re-reading corresponding parts of the KIIs. In line with the objective of this qualitative study, the analysis aimed to showcase the range of different perceptions.

The whole data analysis was an iterative process; transcripts were consulted several times to verify certain answers or reread certain KIIs. Extracts from the interviews were used to highlight key findings. These extracts were further translated from French to English, and attention was paid to reflect the exact statement that was made in the original language.

### 2.5. Ethical Considerations

Ethical approval for the study was received by Burkina Faso’s ethics committee for health sciences (Comité d’Ethique pour la Recherche en Santé; 2019-2-013) prior to any fieldwork activities. Furthermore, the ethics committee of Northwestern and Central Switzerland (Ethikkommission Nordwest- und Zentralschweiz, EKNZ; Req-2018-00386) and the Institutional Review Board of the Swiss TPH approved the study protocol. Every respondent was informed about the study objectives and procedures and provided written informed consent for the study participation. In this frame, participants were asked for permission to be audio-recorded during the KIIs. For the KII sessions, refreshments were organized but no financial compensation was given. All answers were anonymized and treated confidentially during the analysis of the data.

## 3. Results

### 3.1. Study Population

A total of 36 study participants were interviewed at the three different mining sites. On average, one KII lasted around 41 min. The large deviation from the targeted sample size of 54 can be mainly explained by the fact that no one from the Bissa Gold Mine and only one person from the Houndé Gold Mine management participated in the study. Table 2 shows the distribution of study participants per mining site and stakeholder group. Part A of the KIIs was answered by 36 and Part B by 30 participants. Therefore, the tables and reported numbers on the factor ratings show a total of 30 participants.

### 3.2. Perceived Impacts of Mining on Population Health and Health Determinants

The overall assessment of the perceived impacts of gold mining on the population’s health across all stakeholders is depicted in Figure 2. Clear disparities are visible between the three stakeholder groups, with mining officials having the most positive and community leaders having the most negative perceptions. Furthermore, discrepancies are apparent among community leaders themselves. Half (*n* = 7) of the community leaders assessed the overall impact of mining on health as “a bit negative” or “very negative”, while approximately one third (*n* = 5) assessed it as “a bit beneficial or “strongly beneficial”. The range of perceived health impacts of healthcare providers is smaller, ranging from “a bit beneficial” to “a bit negative” (with one exception).

The outcomes of the ratings on health determinants present a similar picture (Figure 3). All determinants studied were unanimously ranked as “improved” by the mining officials. They perceived not a single health determinant as having worsened. Within the community leaders, more determinants of health were ranked as “improved” than “worsened”. About half of the healthcare providers perceived an improvement in “educational institutions” and “employment”. Several respondents, mainly community leaders, did not perceive any positive impacts of the gold mining companies on health or health determinants. In their opinion, the same illnesses prevail as before, and the mining companies have not done anything to improve the communities’ health. One community leader did not associate the population’s health with the mine in either way. He believes that health is a question of God.

#### 3.2.1. Perceived Impacts of Mining on Education, Employment, Women’s Empowerment, and Income Levels

Respondents of the three stakeholder groups predominantly agreed on the positive influence of mining on educational institutions and women’s empowerment. They appreciated the mines’ help in building or refurbishing school buildings and sometimes offering scholarships to very good students. Regarding women’s empowerment, one community leader described the establishment of small enterprises for women to produce soap or weave wool. However, the respondent relativized the mentioned benefit by noting that the soap fabrication was no longer functional. Mining officials also stated that the mining company had financed projects that foster income-generating activities such as the production of spices, soap, wool, soybeans, or the breeding of livestock. Moreover, they stated they would be subsidizing women’s associations that clean up the town or cook on site for the mining staff. A community leader also mentioned the latter. Regarding women’s health, one mining official stated:
*“It’s among our priorities, the women, the youth; so, health is among our priorities in terms of community investments”.*(M1_mining official)

In contrast, a few community leaders and healthcare providers expressed concerns about the closure of artisanal mines due to the industrial mining development, which used to provide a substantial income for women.

When asked about the influence of mining on employment, most healthcare providers (*n* = 7) and mining officials (*n* = 4) felt that it had improved through the employment of many young people at the mine. They believed that those employed by the mine have more financial means to afford healthcare and enough food. Moreover, the mine’s employees often benefit from a health insurance for themselves and their family members, as stated by a mining officer. Unlike the rather positive perception of healthcare providers and mining officials, community leaders’ perceptions of the mines’ impact on employment diverge. Exactly 50% of community leaders rated employment as worsened, whereas 43% said it improved (one participant (7%) said there was no effect). One community leader perceived the youth’s employment as a sign of hope. He assumed that after satisfying their own needs, they will help others improve their lives; hence, an improved well-being will slowly drive and expand throughout the broader community. Similarly, another community leader appreciated the improved living conditions of young people who found a job at the mine but criticized that some of them have already been dismissed, resulting in even worse living conditions than before. Moreover, the population’s disappointment about not being employed by the mine and that the unemployed are now living in despair was emphasized by community leaders:
*If you can’t make something with your hands, that doesn’t work. Before, we sold, and it was bought. But since these two years nothing is bought. They closed the artisanal site that existed and the people no longer have money to buy what they want. So, this is like a disease on us now. When you can’t get money for what you need, that’s a disease. So now it’s not okay.*(M3_community leader)

Surprisingly, healthcare providers’ assessments of income levels (Figure 3) are at odds with their reported improvements in employment and are not due to differences across study sites. A total of 75% of healthcare providers rated income levels to have worsened (*n* = 9). They reported that the loss of access to artisanal mines, increasing prices at the market, and the loss of access to agricultural land due to the resettlement of communities by the mines resulted in less available food and income for the families. Hence, undernutrition was stated as a direct negative health effect as well as less disposable cash for healthcare. Moreover, one mining official revealed that the morals of the people could be affected and result in consequences on the psychological health. Similarly, healthcare providers mentioned difficulties for the communities in adapting to the new living conditions including the loss of important ritual grounds. As stated by one healthcare provider, this led to conflicts and the death of some people at one of the mining sites. Also, 57% of community leaders (*n* = 8) sensed a deterioration of income levels. According to them, the dismantling of artisanal mines exacerbates poverty and consequently results in more under- and/or malnutrition and the inability to afford healthcare. Although a mines’ community manager reported that the community inhabitants’ capacity to pay for healthcare improved, the respondent stated that this was only the momentary situation of the people who were able to keep an income generating activity or obtained compensation payments.

#### 3.2.2. Perceived Impact of Mining on Environmental Determinants of Health

Regarding environmental determinants of health, dust was the most frequently mentioned negative impact at all three mining sites and this across all stakeholders. Adverse health consequences cited ranged from respiratory tract infections, cough, itching, and having a cold to fatiguing or even death. The ratings on different health conditions matched the findings of the open-ended interview part. Many respondents classified respiratory tract infections, and other diseases associated with the respiratory tract have aggravated due to the mining activities.

Moreover, community leaders complained about the noise pollution stemming from the explosions and the roaring mining machinery. Their sleep and peace were also disturbed since mines typically operate around the clock. They also feared future health problems like hearing disabilities. Several community leaders expressed concerns about cracks in their houses and collapsing roofs in connection to the explosions. Healthcare providers did not state noise as a negative impact except for one person in Kongoussi. Officials of one mine acknowledged that they are aware that the mining activities cause noise. However, they relativized it with the facts that the mine is operated underground, measuring devices are set in place and the population does not live near the site.

Also directly linked to the gold extraction are toxic products, such as cyanide. Several healthcare providers and community leaders feared the eventual detrimental health effects of the water, air, and environment contamination by this chemical. One community leader expressed the following perception:
*“There are the tailings of the gold washing with the cyanide, and we are afraid because we do not know if there is infiltration of the subsurface. Will we be able to have clean water to drink in the future in this village. The odors that we smell in the open air, in the long run, what are the diseases that we could suffer from? That means we will die like chickens”.*(M2_community leader)

Regarding artisanal mines, some community leaders and healthcare providers believe their closure may benefit public health because of the lack of safeguards in artisanal mining, specifically against cyanide or mercury poisoning and accidents.

Healthcare providers were especially concerned about the disposal of the mine’s waste products and their potential future health effects. Particularly, healthcare providers criticized the location chosen for the waste disposal grounds right next to the people’s homes instead of behind the hills, as originally planned by one of the mines.

#### 3.2.3. Perceived Impact of Mining on Accidents and Associated Factors

Increased road accidents were another concern of community leaders and especially healthcare providers. Six out of ten health care providers ranked injuries to have aggravated and reasoned it with the augmented number of roadside accidents due primarily to alcoholism. Linked to alcohol and drugs in general, a few respondents also worried about mental health disorders. A doctor appealed to the personal responsibility of the population:
*“The main danger is not the mine itself, but the population! Advise people to drink a little less, to lead a less dangerous life. What we notice most often are accidents that are not the result of the mining company! But of individuals who abuse the alcohol”.*(M1_healthcare provider)

Other cited reasons for the accidents were denser traffic due to population growth and more frequent speeding by the youth. Consequently, community leaders want to sensitize the youth through identification plates and road signs.

### 3.3. Perceived Benefits and Challenges for Health Service Delivery Due to Mining

Mining officials assessed their impact on the delivery of health services mostly as strongly beneficial (*n* = 3), as shown in Figure 4. Also, more than half of community leaders (*n* = 8) and healthcare providers (*n* = 6) perceived impacts to be positive. In total, it can be observed that 18 of 30 respondents perceived rather positive effects on healthcare delivery coming from mining, while only 6 of 30 respondents perceived negative effects. However, it must be noted that certain community leaders had difficulties answering the questions on the mining operations’ impacts on healthcare delivery. They stated that they were unaware of the ongoings at the health centers or who were the benefactors of certain donations. Probably due to similar reasons, community leaders’ statements sometimes contradicted those of the healthcare providers. For instance, one community leader said there was no higher workload at the community health center, whereas a healthcare provider said the opposite.

Appendix B features a summary on the positive and negative perceived impacts on healthcare delivery, categorized according to the focus areas and domains of the WHO’s SARA assessment tool [58]. Regarding the domain health infrastructure, participants reported positively on the additional health centers or the renovation of decrepit health facilities. Furthermore, they mentioned the availability of workforce medical facilities only dedicated to mine staff and dependents, thus minimizing the additional burden on the local public health system. According to the healthcare providers, mine workers consult the public health facilities after an accident, being sent there from the mine, or in cases where they just prefer going to the local, public health facility.

Healthcare providers perceived a higher workload and newly occurring diseases at the health centers and stated the need for a larger and better educated health workforce. In contrast, one healthcare provider believed that the town council had received financial support from the mine that could be used to employ more community health workers, whilst the town council’s leader stated that he had to seek another location for a diagnostic test since the waiting queue at the local health center was too long. On a positive note, one respondent mentioned that a community health center was able to profit from monthly visits by the mine’s doctor. He offered extra consultations for the population which relieved the health workers.

Throughout most of the interviews with community leaders and healthcare providers, respondents highlighted the primary challenge to ensuring healthcare provision in an industrial mining setting, which is the increased strain on the health sector due to higher patient numbers and service utilization. Primarily, patient numbers were said to have increased due to an influx of people looking for work at the mine and the occurrence of new diseases.

Concerning the availability of basic amenities at the health centers, healthcare providers listed several helpful improvements made by the mining companies. Mostly, they appreciated the donation of the ambulances and, hence the faster transportation of patients. Nonetheless, some respondents expressed their wishes for a higher quantity and quality of the donated ambulances. Secondly, community leaders seemed relieved about the construction of boreholes and human-powered water pumps, ensuring clean drinking water access and ameliorating hygienic and sanitation conditions at the health facilities. The respondents also welcomed the donation of mattresses and beds, though some healthcare providers complained that they were unusable since medical beds should not be made of wood and medical mattresses should have a cleanable leather surface.

Lastly, mining companies were said to have helped by organizing or financing sensitization programs of specific diseases such as malaria, HIV/AIDS, or cervical cancer. This is in line with the perception of most community leaders who reported a reduction of malaria due to the sprayings and the distribution of nets. Regarding the toxicity of cyanide, numerous healthcare providers expressed gratitude for the training provided by the mine on the risks associated with cyanide and other corrosive substances, along with the presentation of potential protective measures. Nevertheless, some healthcare providers emphasized that more sensitization programs are needed prior to the start of the project to raise awareness of both industrial and artisanal gold mining risks. To implement the programs, healthcare providers reported that the mines were mostly working in collaboration with an NGO or other unspecified associations, referred to in the interviews as “some associations” or “some NGO”.

## 4. Discussion

This qualitative study aimed to explore the range of perceived impacts of industrial gold mining on the health of affected communities, specific health determinants, and healthcare service provision by different key informants. How the perceived impacts differ between mining officials, community leaders and local healthcare providers was of particular interest.

Respondents perceived the impacts of the mining projects on the health of surrounding communities at various levels. Mining officials reported nearly exclusively positive effects on health, health determinants, and health service delivery, whereas healthcare providers and community leaders considered both health-enhancing and health-deteriorating impacts. Perceived impacts on health service delivery were two-fold. On the one hand, several interventions by the mine to improve people’s health or health services were appreciated by the majority of community representatives, including healthcare providers. On the other hand, interventions implemented by the mines were mostly perceived to be insufficient or unsuitable to reduce the negative impacts resulting from the mining activities. The range of impacts reported are in line with previous studies showing effects on social [25,59], economic [28,60,61], and environmental determinants [19,62,63,64] of health. This finding underlines the necessity of including the wider determinants of health when assessing a project’s potential impacts. Interestingly, mining officials’ high awareness of the mine’s environmental impacts, which could affect health, may be attributed to their primary focus on their mine’s contributions or to their obligation to conduct an environmental impact assessment (EIA) before mine development. Accordingly, they might be reluctant to recognize health concerns as, from their perspective, no health concerns arise if the environmental impacts are managed. In contrast, healthcare providers and community leaders witnessed changes at the community scale or beyond, allowing them to observe and express concerns about environmental and health-related impacts. This disparity in perspectives underscores the need for a strategic-level environmental assessment, such as a Strategic Environmental Assessment [65], which can comprehensively evaluate the interlinkages between the environment and human health within the existing EIA practice. Such an approach could lead to a more holistic understanding of the potential impacts of mining projects.

Furthermore, the differing viewpoints of stakeholders also raise questions about the implications of environmental, social, and governance principles on the mining industry [66]. Acknowledging and addressing these disparities can lead to more responsible and sustainable mining practices that take into account the diverse perspectives of those affected by mining operations [67].

### 4.1. Role of Women in Mining Settings

One of the most satisfying results that emerged from the analysis is the overall perception that women’s empowerment improved. This is consistent with the approach of all three mining companies to align with SDG 5, gender equality. The concerns expressed by some interviewees’ regarding the closure of artisanal mines that provided significant income for women are at odds with these affirmations. Nonetheless, some respondents also pointed out that the closing down of artisanal mines can potentially reduce the exposure to harmful chemicals, thereby contributing to an improvement in community health. This situation underscores the interconnected nature of the SDGs, where the achievement of one goal may sometimes come at the expense of another, which does not lead to a net improvement in the overall SDGs. A possible solution offers the opportunity for large-scale mines to cooperate or coexist with artisanal and small-scale mines (ASM) to secure the jobs for the local communities [23,68]. Instead of closing them down, offering support to formalize the ASM sector could benefit both [69,70]. Also recommended by healthcare providers are sensitization programs to educate people on the health risks of ASM to eliminate misunderstandings from the onset. Similarly, respondents of a study in Ghana expected the mining companies to increase the frequency of health sensitization, promotion, and education programs [36].

Leuenberger et al. (2021), who conducted focus group discussions at the same study sites as part of the HIA4SD project, reported that participating women complained about the loss of traditional medicinal plants which were used for cooking, business, or personal hygiene [13]. The loss of livelihood activities for women was described several times in the previous literature and linked to less disposable cash for the women to spend on food or healthcare for their families [71,72,73,74]. Furthermore, several researchers report on the terrible conditions for women living in mining communities due to sexual transactions as well as commercial sex work, violence, or unwanted displacements [16,71,75]. In contrast, some studies pertained to the positive effects on women’s health of the impacted mining communities [25,76]. For instance, Anja Tolonen [76] used demographic health survey data from eight African countries and revealed that the likelihood for women in mining communities to report a barrier to healthcare is decreased. Knoblauch et al. (2018) found that women living in the vicinity of two mining projects in Zambia and Sierra Leone often had better health indicators than women living in communities further away from the mine [25]. Nevertheless, K. Jenkins [77] critically appraised that women’s perceptions and experiences related to NREP have mostly been left out of impact analyses [39]. In light of the 2030 Agenda for Sustainable Development, this situation is unfortunate as women can play a central role in economic growth and human development [78,79,80,81,82].

Although our findings regarding women’s role at the mining sites are, to a certain extent, positive, they should be considered with caution, keeping in mind that our study group was gender- and occupation-imbalanced. It is noteworthy that, in a separate study implemented under the same framing project [5], we conducted a detailed qualitative analysis of the gendered health impacts of a mining project in northwestern Tanzania [55].

Additionally, some participants’ descriptions raise questions about the sustainability of the interventions aimed at empowering women or improving the living conditions for other vulnerable population groups (such as youth). The mines’ efforts point in the right direction, but it is once again evident that their interventions and measures need to be tested for their cultural appropriateness, effectiveness, and sustainability.

### 4.2. Positive and Negative Impacts on Determinants of Health

Community leaders and healthcare providers mentioned various positive and negative impacts on determinants of health. Overall, neither risks nor benefits prevailed, and no clear tendency was observed across stakeholders, indicating inconclusive opinions. The fact that a mining community comprises several subgroups that have distinct SDH from the very beginning could be a possible explanation for their indecisiveness. For instance, the observed differences in perceptions on the topic of employment or income levels could be related to SDH such as educational attainment or place of residence [83]. This is consistent with the findings of Caxaj et al. (2014), whose participants criticized the division of the society into the “haves” and “have-not” [84]. They referred to the fact that only a chosen few acquire a job at the mines [85]. Other studies made similar observations in the large-scale mining sector, revealing, for instance, the emergence of jealousy [69] or the rupture of community bonds due to divided opinions as to whether mines are beneficial or not [84]. Furthermore, our finding resonates with Michael, who states that the SDH’s imbalanced distribution generates health inequities [86]. This has also been shown in mining settings across SSA, where some population sub-groups have been affected disproportionally and communities perceived an increase in health inequities [11,55,87]. In the current study, differences were mainly associated with a higher affordability of healthcare and thus an improved health status. Although our study did not explicitly assess the impacts on different population subgroups, our findings suggest a stratification of the population into three subgroups: (1) people who were employed by the mine as well as their families (probably highest benefit); (2) people who received compensation payments by the mine (benefitted but only in the short-term); and (3) the rest (also benefited from positive impacts for the public, but no additional benefits like 1 and 2). A fourth group could present the ones who were primarily employed by the mine but lost their job at some point. As suggested by Leuenberger et al. (2019), to better understand health inequities, stratifying project-affected communities into distinctive subgroups already during data collection is important in determining how their health and well-being is differentially impacted by mining activities [41]. Regarding the promise to “leave no one behind”, as proposed by the 2030 Agenda for Sustainable Development or “health equity”, which is one of the guiding principles of HIA, stratification shall be of high importance for future studies.

### 4.3. Impacts of Mining on Healthcare Delivery

Most of the perceived positive impacts of mining were related to healthcare delivery as most interventions by mining companies were aimed at improving the quality of healthcare services already available. Nevertheless, participants regarded the increased pressure on the health sector and its human and medical resources as problematic. Similarly, participants of a Canadian study reported that the emergence of mining had overburdened health services and, in response, suggested building stronger relationships that could benefit both sectors [16].

Our results indicate that the positive impacts on health service delivery prevail, but more research into the topic is needed due to a lack of studies investigating the influence of mining on healthcare services at the local level. Determining if the perceived positive effects could be converted into improved the health status of the affected population would be of interest to further research. Furthermore, including healthcare workers (including community healthcare workers and healthcare providers) in future studies is necessary in order to gain insight into the health of local populations. As highlighted in the current study, their close connection to communities, including the most vulnerable population groups, as well as their knowledge of public health, is essential to future investigations. Still, healthcare providers wished to be able to substantiate their statements with actual health data, which calls for the government’s help in receiving the necessary information technology for acquiring and storing such data. At the district level, some health data are already available from the District Health Information System 2 (DHIS2) [88]. Previous studies combined these quantitative and qualitative data to contextualize the latter [89,90,91]. Data triangulation [51,92] could help to determine if perceived impacts concur with actual health data and broaden our understanding of the situation at the three probed gold mining projects [42].

### 4.4. Untapped Potential of Health Impact Assessment

Consistent with what has been found in previous studies, our results underpin the importance of turning HIA into common practice for large extraction companies [7]. Especially in SSA, the health status of the impacted populations is very prone to changes to the environmental, social, and economic determinants of health caused by the projects. In the HIA approach, these determinants of health are systematically considered. Our results show that several well-intentioned interventions by the mines did not suit the communities’ expectations, demonstrating a need to involve all affected stakeholders equally during decision-making [27,35,36]. Mabey et al. (2020) described community involvement as a requirement for responsible environmental governance in natural resource management [38]. Even more imperative is the fact that community engagement is critical for disease control and for improving the populations’ health [93,94,95,96]. This is particularly concerning regarding the loss of access to medicinal plants and, therefore, traditional treatment methods. Moreover, this poses a risk to the preservation of generations of skills and knowledge.

Further, responses showed that positive and negative impacts on health are not equally distributed across the impacted population: place of residence, resettlement, employment at the mine, or gender may affect the degree of effect, for example [11]. To fully realize their potential of lasting positive development [37], mining companies should not only aim for equitable distribution of positive impacts among people but also strive to ensure the sustainability of these impacts. This appears particularly imperative as the impacts of mine developments on health are often overlooked and the literature has only found negative implications for the local communities living next to closed mining sites [97,98,99,100]. In addition, there is ample scientific evidence of adverse health impacts from industrial mining projects that could and should be used when implementing new projects. For instance, it has been shown several times that compensation payments are mostly an unsuitable approach for the resettlement of people [101,102,103], yet the studied mining companies used compensation payments as the go-to method.

### 4.5. Limitations

It is plausible that several limitations might have influenced the results obtained. First, there is some likelihood of respondent bias among the interviewees. On the one hand, healthcare providers and community leaders occasionally stated needs such as “we need more ambulances than just one”, which could be read as statements with the expectation of receiving future donations. Analogously, study participants of a case study in Ghana by Lawson and Bentil (2013) tried to gain the sympathy of the researchers by over-reporting on certain topics [104]. Moreover, healthcare providers seemed cautious when mentioning the mine’s negative effects. For example, one healthcare provider noted a recent increase in certain diseases but stated that they were unable to definitively attribute them to the mine due to a lack of pre-mining health data. Healthcare providers also seemed to be knowledgeable about the diseases linked to mining impacts, whereas community leaders did not always recognize the link between mining and the clinical conditions. On the other hand, mine officials did not admit any negative impacts of their operations, which could be a sort of social desirability bias because they feared consequences for their business or wanted to reflect well on themselves [105,106].

Second, some conclusions about the perceptions of mining officials should be viewed with caution because mining officials were not available at all three sites studied. Still, their interviews gave important insights for the current study, while not being representative or comparable. The three mining sites were chosen for their similarity and their duration of existence. Yet, the findings presented reflect the unique situation of the settings and the stakeholders’ perceptions, which form an essential contribution to the scarcity of qualitative research about health in mining settings. Finally, the interviewees were not public health or HIA experts and thus might not be aware of all potential health impacts.

## 5. Conclusions

NREPs present great potential to promote sustainable development in project areas. To achieve sustainability, the promotion of population health is fundamental. This qualitative study explored perceptions of the impact on the health and health determinants of the affected communities, as well as healthcare service provision in industrial mining areas.

Mining officials’ perception of predominately positive health impacts represents a potential risk of insufficient acknowledgment of stakeholder concerns and mining-related effects on community health and health determinants. Therefore, research on this topic should be intensified with the ultimate objective of making HIA, or a solid health component within other forms of impact assessment, a requirement for extractive industries in SSA and thus fully untap the promise and potential of HIA. In addition, further research on the impacts of extractive industries should investigate how resources are allocated within affected health systems, delve into partnership arrangements among local actors, and thoroughly investigate gender-specific impacts to better understand the complexities involved. Overall, this study enhances comprehension of the complex interplay between mining operations and health, emphasizing the need for comprehensive assessments, stakeholder involvement, and sustainable practices in order to mitigate negative impacts and promote the well-being of local communities.

## Figures and Tables

**Figure 1 ijerph-20-07167-f001:**
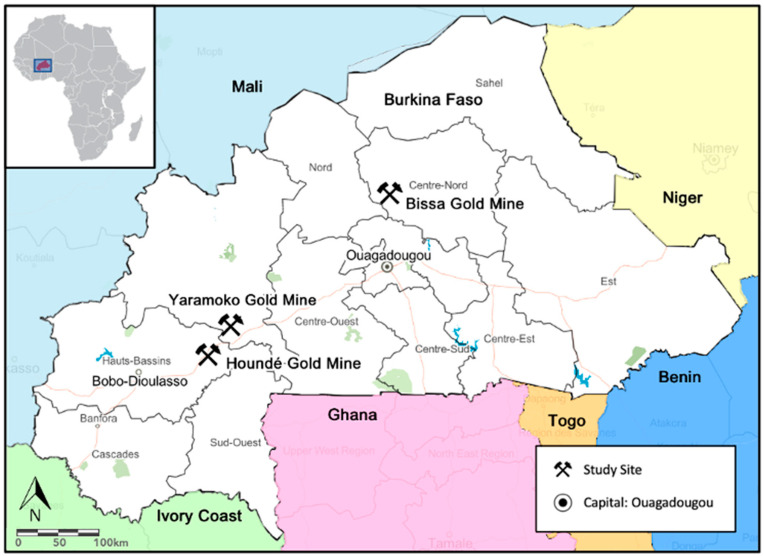
Map of Burkina Faso depicting the location of the three gold mine sites included in the study. (Note: the map was created with ArcGIS Online (2021, Esri Schweiz AG, Zurich, Switzerland), image source: Map data © OpenStreetMap contributors, Map layer by Esri).

**Figure 2 ijerph-20-07167-f002:**
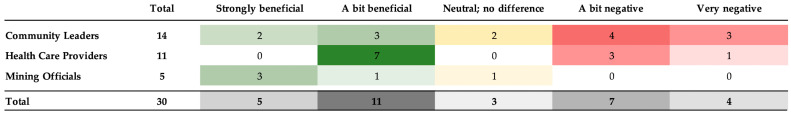
Overall assessment of the perceived impacts of mining on the health of the community. (Note: Numbers sorted by different stakeholders; highest color intensity indicates maximum answer quantity; red = negative, green = beneficial, yellow = neutral).

**Figure 3 ijerph-20-07167-f003:**
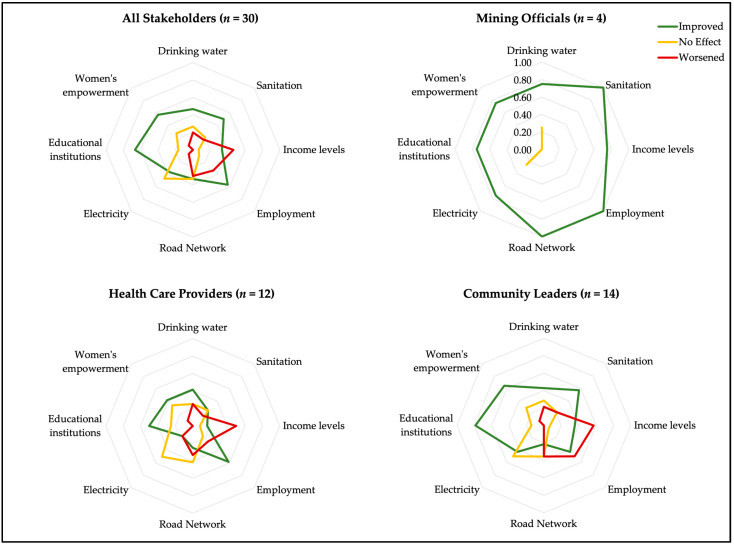
Radar graphs showing the perceived impacts of mining on specific determinants of health categorized by stakeholder group (note: 0 = no participant of a group; 1 = all participants of a group. Some respondents answered “I don’t know”, which has been omitted for analysis).

**Figure 4 ijerph-20-07167-f004:**
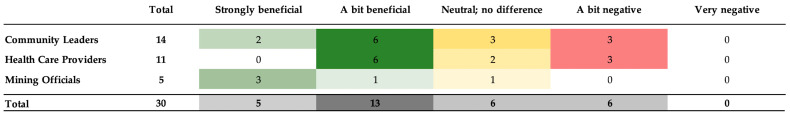
Overall assessment of the perceived impacts of mining on the delivery of health services. (Note: Numbers sorted by different stakeholders; highest color intensity indicates maximum answer quantity; red = negative, green = beneficial, yellow = neutral).

**Table 1 ijerph-20-07167-t001:** Features of the three mining projects in Burkina Faso.

Health District	Kongoussi	Houndé	Bagassi
Mining project	Bissa Gold Mine	Houndé Gold Mine	Yaramoko Gold Mine
Operator (location of corporate office)	Nord Gold SE (Russia)	Endeavour Mining (UK)	Roxgold Inc (Canada)
Commodity	Gold	Gold	Gold
Operational since	2013	2017	2016
Type of mine	Open pit	Open pit	Underground
Population Size (2006)	71,000	77,000	33,000
Data collection	May 2019	March 2019	April/May 2019
Expected mine life	Until 2034	Until 2027	Until 2027

**Table 2 ijerph-20-07167-t002:** Sociodemographic characteristics of study participants.

Mine	Gender	Mining Officials (*n* = 5)	Community Leaders (*n* = 15)	Healthcare Providers (*n* = 16)	Total *n* = 36 (%)	
Bissa Gold Mine	Male	0	4	3	7	10
	Female	0	1	2	3	
Houndé Gold Mine	Male	1	4	5	10	11
	Female	0	1	0	1	
Yaramoko Gold Mine	Male	4	4	6	14	15
	Female	0	1	0	1	
Total	Male	5	12	14	31 (86%)	36 (100%)
	Female	0	3	2	5 (14%)	

## Data Availability

The data presented in this study are available on request from the corresponding author.

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
