# Peer review of "Exploring the Impact of Mining on Community Health and Health Service Delivery: Perceptions of Key Informants Involved in Gold Mining Communities in Burkina Faso"

_ijerph, 2023, doi:10.3390/ijerph20247167_

Round 1
Reviewer 1 Report
Comments and Suggestions for Authors
There are many thought-provoking aspects to this paper. However, based on my review, it is not suitable for publication in IJERPH.
The effort itself was impressive in scope: engage a multi-stakeholder group across three Burkina Faso mining communities to collect perceptions of mines on health. However, these perceptions were then untested.
The study engaged a multi-stakeholder group of public, private, and health-care entities, with health as a common denominator in the KIIs, which facilitates comparisons across different perspectives. The focus of KIIs on health demonstrates a heightened awareness of the interplay between contaminants and SDOHs on health. The documentation of KII responses and organization of qualitative results was also sufficient.
I believe Appendix B is where the ultimate conclusion to reject publication in IJERPH is determined. This is a table of perceived positive and negative health-related effects of mining without any experimental data to substantiate them to any informative degree. While the study acknowledges its qualitative nature, these survey results should still be tested against some type of metric, measurement, or quantification to be minimally considered for publication in IJERPH.
Results of the KIIs (Appendix B) and their relation to Burkina Faso mining may be of interest, yet the perceptions should, at a minimum, be compared to available data, or tested with additional experimentation and measurement collection. This would provide a basis upon which to test perception validity, relative risks/benefit (perceived vs. actual), and metrics for follow-up analysis (e.g., before/after intervention strategies). This does not suggest that the perceived risks/benefits are invalid, but if they are not tested, then this research lacks the experimental and theoretical rigor required for publication. I can only assume it took a tremendous effort to get this far and attain these results and would encourage the authors to consider publication elsewhere or dive more deeply into investigating the perceptions to be considered for publication in IJERPH.
Reviewer 2 Report
Comments and Suggestions for Authors
- Thank you for the opportunity to review your manuscript. It is an important area of study in sustainability and environmental impact. As you identify, the impact of mining on local social and economic structure and the impact on health/wellbeing are well suited to qualitative methods. The insights from people active in the community can add important understanding of the lived experiences of people who are working within the contexts. It is with consideration that I propose a reworking of the paper to address some of this potential for meaningful additions of insight in the field. The nuances missing that would be of value include- the politics of allocation of resources, the context of closure of the artisanal site and the reallocation of the work of women ( what local knowledge is lost and what are the impacts on sustainability?), the lack of transparency ( or presence) of local consultation which would be expected in claims of corporate social responsibility. I suggest some possible areas for addressing this in my comments below and in the annotated transcript attached. Further insight into the data you have collected by more inclusion of granular analysis of these questions would make this a more valuable contribution to the literature.
- The manuscript is clear, relevant for the field and presented in a well-structured manner. A generalised gap is identified but seems post hoc. The manuscript begins promising more insight into health allocations and collaboration, and the findings address some of this, but not with the depth expected of a qualitative study. More insight into the areas of collaboration between management, health centres and local leaders would assist in this. What did they say about each other? Why were substandard resources (ambulances, hospital beds, etc) provided, and who made these decisions? What are the mechanisms that are currently triggering these issues, and are there any indications of how they can be overcome?
- The cited references are mostly recent publications (within the last 5 years) and relevant? There are a high number of self-references ( Winkler 10 instances, Leuenberger 9 instances) however this may be due to eminence in the field.
- The manuscript is generally scientifically sound experimental design appropriate to test the hypothesis. The hypothesis, however, is not suggesting something that is new to the field. Some of the cited literature, which was published later than this survey that are done by one of the authors at the same sites (Leuenberger) indicate a nuanced understanding of the level of inequity of women’s representation and health outcomes, yet in this study, design there is a lack of recognition of that nuance. The manuscript highlights strength in being a qualitative survey, but there is little insight into the complexity of representation ( or lack of representation) in the findings. Tables, in particular, are presented with a lack of the granularity that would give insight into the differences in gender and location ( open versus closed pit mining). It is only identified in the later findings that managers were not representing all the sites, which has impact on the way the data is read. Reconfiguration of the tables or removal of the tables is recommended and they represent an unbalanced quantitative overview of a qualitative study. I have annotated the manuscript with a number of questions and considerations that might be asked of the transcript data. The strength of the methods used are not equally applied in analysis, and as a result the manuscript largely reasserts the findings outlined in the existing literature. The real opportunities to glean insights into local challenges in resource allocation, local conceptions of wellbeing and local health practices are lost- alluded to in passing, but without adequate inclusion of data or insight to indicate how they might be understood through the discussion.
- The conclusions are somewhat consistent with the evidence and arguments presented but weigh heavily on the extensive limitations that seem foreseeable in the original design. The disruption of exisiting economic structures, the closure of the artisanal mine and the redirection of the labour of women, the public health concerns around pulmonary health, the importance of unsustainable cultural and economic practices/ environmental hazards around fixed term mining are all recognised in the literature over decades now, and in the design and analysis of qualitative research in this field, it would be expected that processes would be put in place to gather this information and ensure equity of voice in a population of research participants.

Reviewer 3 Report
Comments and Suggestions for Authors
The theoretical foundation should be prominent.
Comments on the Quality of English LanguageThe language proficiency of this article is acceptable.
Reviewer 4 Report
Comments and Suggestions for Authors
Interesting read. Clear and precise description of the problem statement, research questions that seek answers and the methodologies. Minor issue with the reference errors in Sections 2.1, 3.2 and 3.4.
In Section 3.3.1, one of the perceived impacts among the community leaders was the loss of jobs due to the shut down of artisanal mining sites giving way to those of industrial scale. It is worth to also consider or to mention what could be the health impacts if the artisanal mining sites would have been continued; and whether the loss of job is a net loss or otherwise; and the side effects of mining at industrial scale while providing some benefits. These considerations would be useful for discussion on the potentially conflicting SDGs achievements where an achievement of one SDG may be at the expense of another SDG.
Another important point that needs to be highlighted is that, the mine officers provided their opinion based on their respective mine's contribution while the health officer and the community leaders witnessed the change at the community scale or larger. This may naturally lead to the potential recommendation of the strategic level of environmental assessment (e.g. Strategic Environment Assessment) and also the interlink between environment and human health in the existing EIA practice. Another suggestion would be the implications of ESG on the practice of mining industry.
Round 2
Reviewer 1 Report
Comments and Suggestions for Authors
I appreciate the authors' patience in replying to my comments. I remain reserved in my decision but will defer to the editor and other reviewers with respect to acceptance or rejection. I will quickly reiterate my main concern and then follow up with a couple minor comments to consider.
As mentioned, I believe that conducting these KIIs represented a significant undertaking -- developing the interviews, gathering the approvals, traveling to the sites, engaging translators and stakeholders, documenting and classifying responses, etc. It is difficult enough to do this locally with willing stakeholders. However, asking about perceived risks/benefits represents only the first step of what is typically considered (at least in the United States) a Health Impact Assessment (HIA). In various ways, I have performed this step many times with stakeholder groups and always considered it only the beginning. My concern with the paper is that, as detailed and rigorous as this survey was, and as topical and newsworthy about African gold mines, it goes no further than only documenting these perceptions. The low representation of mining officials is not a critical flaw, but does lessen the impact of the results. For these reasons, I am uncertain whether this should stand alone as a publication and will defer to the others on that. For example, I would wait until whatever the next step is, is performed. Then couple this scoping with that implementation/follow-up and report on it as a whole.
Here are a couple comments to consider based on a decision to revise/resubmit:
Kindly define health officers -- are these community residents employed by the mines? Or are they government or non-profits? What is their relationship to the others?
HIA context -- please clarify what is meant by health impact assessment (HIA). HIA methods are not universal. First, please define HIA and any HIA steps and then describe which part of the HIA process your work accomplishes. Second, please consider the Minimum Elements and Practice Standards for HIA as presented by the Society of Practitioners of HIA (SOPHIA). According to these steps, your work accomplished the Scoping stage. After this would follow Assessment (e.g., data collection), Recommendations (e.g., interventions), Reporting (e.g., back to stakeholders, and Evaluation/Monitoring (e.g., of project itself and its effectiveness). In my previous review, I was looking for something related to these subsequent steps as I do not typically consider Scoping to be sufficient for publication. However, I may not be current on my literature-based understanding of these surveys.
Thank you and good luck!
HIA Minimum Elements and Practice Standards:
Reviewer 2 Report
Comments and Suggestions for Authors
Thank you for applying such granular attention to my feedback. I will look forward to reading the other papers from this project. I am happy for publication to proceed.
Author Response
Response: Thank you for acknowledging our effort and the positive feedback.Reviewer 4 Report
Comments and Suggestions for Authors
Authors have made adequate response to comments.
Author Response
Response: Thank you for acknowledging our effort and the positive feedback.